# The Mode of Action of *Bacillus* Species against *Fusarium graminearum*, Tools for Investigation, and Future Prospects

**DOI:** 10.3390/toxins11100606

**Published:** 2019-10-18

**Authors:** Khayalethu Ntushelo, Lesiba Klaas Ledwaba, Molemi Evelyn Rauwane, Oluwafemi Ayodeji Adebo, Patrick Berka Njobeh

**Affiliations:** 1Department of Agriculture and Animal Health, Science Campus, University of South Africa, Corner Christiaan De Wet and Pioneer Avenue, Private Bag X6, Florida 1709, Guateng, South Africa; ledwabal@arc.agric.za (L.K.L.); rauwaneme@gmail.com (M.E.R.); 2Agricultural Research Council-Vegetable and Ornamental Plants, Private Bag X293, Pretoria 0001, Tshwane, South Africa; 3Department of Biotechnology and Food Technology, University of Johannesburg, Corner Siemert and Louisa Street, Doornfontein 2028, Gauteng, South Africa; oadebo@uj.ac.za (O.A.A.); pnjobeh@uj.ac.za (P.B.N.)

**Keywords:** *Bacillus*, *Fusarium graminearum*, antagonism, mode of action

## Abstract

*Fusarium graminearum* is a pervasive plant pathogenic fungal species. Biological control agents employ various strategies to weaken their targets, as shown by *Bacillus* species, which adopt various mechanisms, including the production of bioactive compounds, to inhibit the growth of *F. graminearum*. Various efforts to uncover the antagonistic mechanisms of *Bacillus* against *F. graminearum* have been undertaken and have yielded a plethora of data available in the current literature. This perspective article attempts to provide a unified record of these interesting findings. The authors provide background knowledge on the use of *Bacillus* as a biocontrol agent as well as details on techniques and tools for studying the antagonistic mechanism of *Bacillus* against *F. graminearum*. Emphasizing its potential as a future biological control agent with extensive use, the authors encourage future studies on *Bacillus* as a useful antagonist of *F. graminearum* and other plant pathogens. It is also recommended to take advantage of the newly invented analytical platforms for studying biochemical processes to understand the mechanism of action of *Bacillus* against plant pathogens in general.

## 1. Introduction

Biotic stresses such as plant pests and pathogens are the major factors threatening global crop production. Proliferation in plants of these pathogens can cause devastating epidemics, which can cause severe food shortages especially in countries with limited resources. The current state of crop losses due to pathogenic diseases is alarming, with an estimated 8–40% of crop yield losses caused by plant pathogens worldwide [1,2]. One of such prominent pathogens of health and economic importance is *Fusarium graminearum.* This fungus is a known causative agent of Fusarium head blight (FHB), which is an economically important cereal crop disease that accounts for worldwide losses estimated between 20 and 100% [3,4,5,6]. According to Dean et al. [7], *F. graminearum* is ranked the fourth most important plant fungal pathogen, on the basis of its scientific and economic importance. This filamentous ascomycete infects floral tissues of cereal plants and contaminates food grains [7]. Infection is associated with premature bleaching symptoms, which mainly reduce grain quality and, less often, yield [3,7,8,9].

In addition to grain quality reduction, *F. graminearum* also produces various types of mycotoxins, which if ingested in huge amounts, cause various toxicoses in animals and humans [9]. The major mycotoxin produced by *F. graminearum* is deoxynivalenol (DON) together with other mycotoxins including the trichothecene nivalenol (NIV) and its derivatives, 3- and 15-acetyldeoxynivalenol (3-ADON and 15-ADON). These mycotoxins are reported to contaminate grain food products, thereby posing a threat to humans and animals by causing neurological disorders and immunosuppression [10,11,12] amongst other dysfunctions. However, these health complications vary from one animal species to the other and according to several factors such as trichothecene type, level, and route of exposure. This assembled body of evidence justifies the need for the biocontrol of *F. graminearum* in several foodstuffs [12]. In the past three decades, control strategies against this devastating plant pathogen have been based solely on fungicide application, which has resulted in long-term undesirable environmental pollution [13]. Herbicides and insecticides have also been used over the years to suppress the activity of this pathogenic microorganism causing FHB, amongst other diseases, in crops. Coupled with fungicides are various control practices such as sanitation, good agricultural practices, as well as the use of resistant cultivars. With the increase in awareness of the danger of chemical control applications, fungicides are beginning to take a back foot, with the use of biocontrol products being exploited.

With the increased desire for environmental friendliness and sustainability, the biocontrol of pathogens is equally receiving attention. Biocontrol is defined as the use of natural products and living organisms to suppress pathogen populations. The use of biocontrol agents either as an alternative to other forms of plant disease control or as a supplement has attracted worldwide attention to be included in an integrated pathogen management strategy in various food systems. However, to prevent an irrational selection of plant pathogen antagonists to be adopted as commercial products, the modes of the antagonists’ activities and effects need to be fully understood. Bacterial antagonists are commonly used, and many of them belong to the genus *Bacillus* [14].

In this perspective manuscript, we summarize the current knowledge about the mode of action of *Bacillus* species against the pervasive plant pathogen *F. graminearum*. Background information about *Bacillus* is provided, the antagonism of *Bacillus* and its mode of action, tools and techniques to uncover the mechanisms of the antagonism are described, and future prospects are presented.

## 2. Overview of *Bacillus* Species as a Protective Agent against Pathogens

*Bacillus* is one of the largest genera of bacteria that produce aerobically dormant endospores under diverse growth conditions [15]. Species belonging to this genus can play a role as human pathogens, whilst others promote plant health and development [16]. Due to their different genetic characteristics, *Bacillus* species are ideal candidates as biocontrol agents. *Bacillus* species play a role as bacterial antagonists to pathogens due to their ability to reproduce actively and their resistance to unfavorable environmental conditions [14]. The species’ antagonistic activities are associated with the production of metabolites with antibiotic properties [17]. Particularly, volatile metabolites produced by these microorganisms also play an important role in the activation of plant defense mechanisms by triggering induced systemic resistance (ISR) in plants [18]. In addition, plant host defense responses can also be activated during the production of metabolites by *Bacillus* species [19]. As documented in the literature, *Bacillus* spp. also directly antagonize fungal pathogens by competing and depriving them of essential nutrients, by producing fungitoxic compounds, and by inducing systemic acquired resistance in plants [20,21,22,23].

A wide range of pathogenic microorganisms have been controlled using *Bacillus*-based biocontrol agents [17,24,25,26]. Several disease control products produced from various strains have also been registered and are commercially available. A broad spectrum of resistance mechanisms against plant diseases have been reported to be induced by *Bacillus* strains in many studies [17,27]. Furthermore, the activity of other *Bacillus* strains was also investigated in different crops and found to be effective against various fungal plant pathogens and diseases, including *Fusarium* wilt in tomato [28] as well as FHB in wheat and barley [19,27].

### 2.1. Biological Activity of Bacillus in General and Against F. graminearum

*Bacillus* species can produce different antimicrobial substances that confer protection and act as biological agents [29]. Such substances include subtilin [30], bacilysin [31], mycobacillin [32], bacillomycin [33,34], mycosubtilin [35,36], iturins, fengycins, and surfactins [37]. These substances have been reported to exert antibacterial and/or antifungal activities against pathogenic microorganisms [17,19,26,27,28,29,30,31,32,33,34,35,36,37,38]. As noted in the literature, among these antimicrobial substances produced by *Bacillus*, the most studied with regard to *F. graminearum* are surfactin, fengycin, and iturin. For this reason, the literature reported herein focuses on these three *Bacillus*-produced antimicrobial agents.

The antagonism of these antimicrobial substances has been reported against *F. graminearum* [26,27], *Fusarium oxysporum* [39,40], *Fusarium solani*, and *Rhizoctonia solani* [40], amongst many other plant pathogenic fungi. In a study by Földes and colleagues [29], antagonistic compounds produced by *Bacillus subtilis* IFS-01 exhibited antimicrobial effects against phytopathogenic, food-borne, and spoilage microorganisms. In an agar diffusion assay, some of the filamentous fungi and yeasts tested showed no visible growth within the inhibition zone (about 10 mm from the colony) due to the antagonistic effect of *B. subtilis* IFS-01. These findings confirmed the biological control ability of this *Bacillus* strain against these fungi and yeasts. The iturin family of the lipopeptides produced by *Bacillus amyloliquefaciens* PPCB044 strain showed antagonism against pathogenic fungi from seven citrus plants during postharvest [41]. All the fungal pathogens were deterred by the *B. amyloliquefaciens* PPCB004 strain, as the strain produced compounds related to iturin A, fengycin, and surfactin [41]. Similar results were also described by Gong et al. [26], who reported the antagonism of iturin A and plipastatin A from *B. amyloliquefaciens* S76-3 in wheat inoculated with *F. graminearum*. The data obtained from both the growth chamber and the field plot assays revealed a strong antagonistic activity of strain S76-3 against the growth and development of *F. graminearum*. Iturin A killed the conidia at the minimal inhibitory concentration of 50 µg/mL, while plipastatin A exhibited a strong fungal activity at 100 µg/mL.

Zalila-Kolsi et al. [19] studied the FZB42 strain of *B. amyloliquefaciens* and found that the commercial bacterial strain produces the lipopeptide bacillomycin D, which contributes to its antimicrobial activity. Bacillomycin D showed a strong antagonism against *F. graminearum* at 30 µg/mL, which is its 50% effective concentration. The plasma membrane morphology and cell wall of *F. graminearum* were affected by bacillomycin D, while inducing the accumulation of reactive oxygen species (ROS) [19]. Furthermore, this lipopeptide caused cell death of the tested *F. graminearum.* Lipopeptide-type compounds from the iturin, fengycin, and surfactin families, synthesized by various strains of *Bacillus*, effectively suppressed the growth of pathogenic microorganisms [26,27,28,39,40,41]. These lipopeptides have different residues at specific positions but consist of variants with the same peptide length. Molecules of the iturin lipopeptide family are linked to a β-amino fatty acid of variable length (C_14_–C_17_), those of the surfactin family to a β-hydroxyl fatty acid (C_12_–C_16_), while fengycin decapeptides are linked to a β-hydroxyl fatty acid chain (C_14_–C_18_) [42]. These nonribosomal peptide synthetase-mediated compounds are surface-active and have emulsifying and foaming properties and haemolytic activity [43,44,45].

Different strains of *Bacillus* produce different groups of lipopeptides [46], and their role in suppressing/controlling plant pathogens may vary. Similar lipopeptides produced by various *Bacillus* strains can also suppress and control other pathogens of economic importance. In a study by Guo et al. [47], the antagonistic effect of the *B. subtilis* strain NCD-2, a fengycin-deficient mutant, was strong against *R. solani* in vitro and suppressed cotton damping-off disease in vivo. In addition, *B. amyloliquefacien* CM-2 and T-5 showed their antagonistic activities against the bacterium *Ralstonia solanacearum* in tomato [28]. The disease incidences were reduced by over 70% by both strains in comparison to the control. On the other hand, crude lipopeptide extracts of *B. amyloliquefaciens* SS-12.6 successfully suppressed leaf spot disease severity on sugar beet plants [48]. These studies showed significant antagonism of the various *Bacillus* strains against various pathogens. Many studies have reported the success of *Bacillus* as a biological control agent against *F. graminearum* in various crops and diseases. However, the potential of these biocontrol agents has not been fully exploited to control other pathogens. Therefore, different strains of *Bacillus* species should be studied further as potential biocontrol agents against other pathogenic microorganisms.

### 2.2. Surfactins, Fengycins, and Iturins in Bacillus Species

The production of surfactins, fengycins, and iturins by various strains of *B. subtilis* has been reported by numerous researchers [49,50,51,52,53,54,55,56,57,58], and a crude lipopeptide mixture of the supernatant of *B. subtilis* was once found to contain these polypeptides [59]. The main congener structures of these cyclic lipopeptide families are shown in Figure 1. Among the most studied in the surfactin family are surfactin linchenysin, pumilacidin WH1, and fungin; for the iturin family, the various iturin isomers—bacillomycins, mycosubtilin—are the best known, while for fengycin, the main compounds are feycin, plipastatin, and agrastatin 1 [60]. An overview of the activity of these three lipopeptides against fungi, with emphasis on *F. graminearum*, is provided in the following sections of this review.

#### 2.2.1. Surfactins

Surfactins are natural lipopeptides that have been reported to possess antifungal activity [42,61]. They include β-hydoxy hepta cyclic depsipeptides with possibile alanine, valine, leucine, or isoleucine amino acid variations at positions 2, 4, and 7 in the cyclic depsipeptide moiety and C_13_ to C_16_ variation in the β-hydroxy fatty acid chains [62,63,64]. Surfactin is amphiphilic, with a polar amino acid head and a hydrocarbon chain. This molecular structure makes surfactin a strong biosurfactant, which is at the basis of its antifungal properties. It is assumed that its antibiotic properties are due to its ability to produce selective cationic channels in the membrane phospholipid bilayer [65]. Several studies have been conducted to determine the effect of surfactin on fungi. Qi et al. [66] found a new surfactin, WH1fungin (Figure 2), which induces apoptosis in fungal cells. The same surfactin has also been reported in other studies as an oral immunoadjuvant that could be used for the development of vaccines [67,68]. Surfactin was also found to be effective against the plant pathogenic fungus *Colletotrichum gloeosporiodes* [57]. Another surfactin, Leu^7^-surfactin, produced by *Bacillus mojavensis,* was found to be effective against *Fusarium verticillioides* [69]. A similar inhibitory activity of surfactin was discovered against *F. graminearum* [17], *F. oxysporum* [70], and *Fusarium moniliforme* (presently *F. verticillioides*) [71]. This effect on *F. graminearum* can be culture condition-dependent [17,19], with iron concentration being the most important determinant [19].

#### 2.2.2. Fengycin

The antimicrobial activity of *Bacillus*-produced lipopeptides is based on their chemistry. This is also the case with fengycin, which is a cyclic lipodecapeptide that contains a β-hydroxy fatty acid with a side chain consisting of 16–19 carbon atoms [73]. Fengycin is particularly active against filamentous fungi and inhibits the functions of the enzymes phospholipase A2 and aromatase [73]. It has various isoforms, which differ in length and branching of the β-hydroxy fatty acid moiety, as well as in the amino-acid composition of the peptide ring [50]. For instance, position 6 d-alanine (as in fengycin A) can be replaced by d-valine (as in fengycin B) [73,74]. Fengycin A presents 1 d-Ala, 1 l-Ile, 1 l-Pro, 1 d-allo-Thr, 3 l-Glx, 1 d-Tyr, 1 l-Tyr, 1 d-Orn, whereas in fengyicn B, d-Ala is replaced by d-Val. 

Fengycin affects the integrity of biological membranes in a molar-ratio-dependent manner. The effects of fengycin on biological membranes depend on the concentration, but ultimately high concentrations completely disrupt membranes [75]. Fengycins are elicitors of plant defense [76] and have been found to be effective against many fungi including *Magnaporthe grisea* [77], *Plasmodiophora brassicae* [78], *Botryosphaeria dothidea* [79], *C. gloeosporiodes* [57], and a number of other fungi [80]. A cluster of fengycin homologues were found to be effective against *F. verticillioides* [80], *F. solani* [81], *F. solani* f. sp. *radicicola* [80], *F. oxysporum* [25,39], *F. oxysporum* f. sp. *spinaciae* [27], fumonisin production by *F. verticillioides* [82] and proliferation of *F. graminearum* [17,27,80,83,84,85]. On *F. graminearum*, fengycin causes structural deformations of the hyphae and suppresses in planta proliferation and mycotoxin production [27,84], permeabilization of hyphae [85], and in planta arrest of ear rot development of maize [83]. The study of Liu et al. [86] also revealed that fengycin could block the growth of *F. graminearum*, disrupt cell membrane structure increasing permeability, and create primary lesions in the membrane of fungal cells, thus compromising cell integrity. While the efficacy of fengycin cannot be disputed, its effect on *F. graminearum* can be concentration-dependent [80,86].

#### 2.2.3. Iturin

Iturins exhibit strong fungitoxic properties by forming ion-conducting pores upon contact with fungal membranes. These amphiphilic compounds possess a heptapeptide backbone connected to a C_13_-to-C_17_ β-amino fatty acid chain [56,87]. Iturins vary in structure, their differences consisting in the type of amino acid residues and in the length and branching of the fatty acid chain. Some examples include iturins A, C, D, and E, bacillomycins D, F, and L, bacillopeptin, and mycosubtilin, all of which are arranged in an lddlldl configurational sequence [88]. Length and fatty acid chain branching heterogeneity is clearly demonstrated by iturin A, which has up to 8 isomers with between the 10 to 14 carbons and branching with *n*-, *iso*-, or *anteiso* configurations of the fatty acid chain [89]. Members of the iturin family bacillomycin and bacillopeptin have different amino acids at the third, fourth, and fifth positions. Mycosubtilin, a *B. subtilis*-produced iturin family member, targets, through its sterol group, ergosterol present in the membranes of sensitive fungi [90]. Bacillomycin L is presumed to act by inducing membrane permeabilization and disruption, as well as by targeting intercellular structrues [91]. Iturins have been found to be effective against a number of plant pathogenic fungi, which include *Botrytis*, *Penicillium, Monilinia* [92], *R. solani* [93], *Colletotricum* [94], *F. oxysporum* [95,96,97], and *F. graminearum* [19,26].

On *F. graminearum,* iturin causes morphological distortions in conidia and hyphae and severe damage to the plasma membrane, which lead to leakage of the cell contents [26]. Figure 3 illustrates the effects of iturin on *F. graminearum* conidia. Co-cultured with *Bacillus, F. graminearum* is not able to decrease the germination ability of wheat seed [19].

## 3. Techniques Applied to Establish Potential Modes of Action of *Bacillus* against *Fusarium graminearum*

The effect of an organism or a substance against the growth of a target organism is traditionally studied by means of bioassays. In a bioassay, the organism is grown in the presence of the antagonist, and its growth monitored over time in comparison to that of an experimental control. Characteristically, a zone of growth inhibition is formed around the inhibited microbe. Various bioassays have been conducted to assess the effect of *B. subtilis* on the growth of *F. graminearum*. Notable is the study of Zhao et al. [27], which clearly demonstrated an antagonism of *Bacillus* against *F. graminearum*, whose mechanism still remains not fully elucidated. If a polypeptide is suspected to be a growth deterrent against target microorganisms, genes (their presence or relative expression) which code for the polypeptide can be detected in the growth culture by means of the polymerase chain reaction (PCR) technique. This was the case in the studies of Arrebola et al. [41], Velho et al. [98], and He et al. [99].

The questions needing answers would then be: What are the antagonistic compounds and how do these antagonistic compounds inhibit growth? Studies based on bioassays analyze the growth medium in which the antagonistic microbe and its target are grown. As part of the biochemical analysis, this growth medium is compared with a control growth medium, and inhibitory compounds are detected. Detection is done using techniques such as liquid chromatography–mass spectrometry (LC–MS). Examples of these studies are those which were conducted to detect and/or analyze surfactin, fengycin, and iturin produced by *Bacillus* against various plant pathogenic fungi [77,100,101,102,103,104,105,106,107]. Two initial scenarios may require this type of testing. The first is when the presence of a specific compound responsible for the antagonistic effect is supposed. This is a targeted analysis, which seeks to confirm the presence of the ‘suspected’ compound. Alternatively, if the presence of a specific molecule is not presumed, an untargeted analysis to assess culture conditions in comparison to the control is performed. A target analysis follows this untargeted analysis. Studying the antagonistic effect of *Bacillus* against *F. graminearum* for the protection of durum wheat, Zalisa-Kolsi et al. [19] performed an in vitro bioassay, which was followed by an in planta growth inhibitory test. Similarly, in studying the effect of three *Bacillus* strains against *Fusarium*, Dunlap et al. [17] followed a radial diffusion assay with analysis of candidate lipopeptides using high-performance liquid chromatography (HPLC) and a matrix-assisted laser desorption/ionization time-of-flight (MALDI-TOF) system. Zhao et al. [27] performed a similar experiment and discovered an antagonistic effect of *B. subtilis* strain SG6 on *F. graminearum*, as many other similar studies [26,108].

## 4. Tools for the Detection of Surfactin, Fengycin, and Iturin Genes in *Bacillus* Strains and Culture with Biological Activity against *Fusarium graminearum*

Genomic analysis of *Bacillus* has shown that these bacteria possess genes which code for metabolites associated with biological control [38,109,110,111,112,113]. Genetic information made available by genomic sequencing has led to a better understanding of *Bacillus* biocontrol features. Chen et al. [112] characterized the genome of the *Bacillus velezensis* LM2303 strain, known for its strong biocontrol potential against *F. graminearum*. This strain presented the largest number of biocontrol genes and gene clusters when compared with strains studied earlier. Thirteen biosynthetic gene clusters associated with biocontrol activity were identified using an integrated approach of genome mining and chemical analysis, including the three antifungal metabolites fengycin B, iturin A, and surfactin A [112]. Another strain, *B. velezensis* LM2303, which has antimicrobial activity against *F. graminearum* in addition to other plant pathogenic fungi, also presented a plethora of genes encoding antimicrobial compounds. These findings demonstrated the value of genomic analysis in both biocontrol strain characterization and understanding of the basis of biocontrol activity. A plethora of co-culturing studies have utilized PCR to detect genes involved in biological control in culture, to later identify the basis of their biological control activity. However, biological control genes are sometimes detected in pure *Bacillus* strains undergoing characterization [46]. The study by Adeniji et al. [46] analyzed seven isolates of *Bacillus* with bio-suppressive effects against *F. graminearum* and found them to have valuable gene clusters encoding biocontrol agents. The fingerprint of the combination of genes detected by PCR indicates that strain differentiation and selection are important to identify the strain demonstrating the highest antimicrobial activity as a candidate biocontrol agent. Studies to identify surfactin, fengycin, and iturin in culture are routinely carried out and have uncovered a myriad of antimicrobial substances able to act against plant pathogenic fungi, including *F. graminearum*. These studies make use of combined chromatography and mass spectrometry to identify the compounds which have antagonistic activity. Using reverse-phase high-performance liquid chromatography and electrospray ionization mass spectrometry (RP-HPLC/ESI–MS) analyses, Gong et al. [26] identified iturin and surfactin in a culture of *B. amyloliquefaciens* isolated from wheat infected with *F. graminearum*. Further characterization of iturin showed that it causes leakage and/or inactivation of *F. graminearum* cellular contents. Using thin-layer chromatography–bioautography, Lee et al. [96] identified iturin A in a butanol extract of a culture of *B. amyloliquefaciens* strain DA12, which was found to be active against *F. graminearum*. The same study also attributed this activity to volatile heptanones, some of which were detected using gas chromatography–mass spectrometry (GC–MS). A similar study was performed using ultra-high-performance liquid chromatography coupled with mass spectrometry (UHPLC–MS) to confirm the presence of fengycin B, iturin A, and surfactin A in *B*. *velezensis* [112]. Also, the study of Adeniji et al. [46] used electrospray ionization–quadrupole mass spectrometry (ESI–Q-ToF-MS) to detect surfactin, fengycin, and iturin in the *F. graminearum*-supressing *B. velezensis* strain NWUMFkBS10.5. The power of these analytical techniques lies on their sensitivity and accuracy of detection, and their application is critical for, amongst other things, the detection of toxins in food to ensure compliance with food safety standards based on critical threshold values. Moreover, their application to detect bioactive components of *Bacillus* against *F. graminearum* is particularly relevant.

## 5. Future Prospects and Conclusions

The evidence that *Bacillus* species can act as biocontrol agents against *F. graminearum* encourages the exploitation of *Bacillus* in crop protection and their potential use for organic farming to supplement the despised control measures that pose various environmental hazards and health risks. Ideally, their use may completely replace the current strategies for the control of *F. graminearum* in wheat and other crops. This is supported by various studies conducted to assess the suitability of *Bacillus* to control wheat diseases, in particular FHB. The biofungicide, *B. subtilis* strain QST 713 suspension concentrate (Serenade^®^ASO) was tested against yellow rust in wheat and showed promising applicability for the control of this fungal infection. However, control tests proved that this biofungicide can be more effective as part of an integrated control strategy than as a standalone remedy [114]. Further work is, therefore, necessary to design an integrated control strategy which utilizes Serenade^®^ASO together with other organic disease control methods. *B. amyloliquefaciens* CC09 was also reported to have great potential as a biocontrol agent for wheat powdery mildew [115]. The same CC09 strain was found to be effective against take-all disease caused by *Gaeumannomyces graminis* and against a myriad of symptoms caused by *Bipolaris sorokiniana*. This strain effectively colonized the wheat tissue and was found to express genes encoding iturin A synthetase, thereby gaining the name “potential vaccine” [116]. Through its ability to also form spores, *Bacillus* can be an effective biological control agent against *F. graminearum* in wheat. With spore formation, *Bacillus* can overwinter and protect wheat against FHB over several growing seasons. Although the use of biocontrol agents must be extensively tested, ensuring they have a reasonable shelf life, compatibility with other treatments and affordability must be ascertained. Such is not the case with *Bacillus*, which seems to have passed many of these hurdles to become an effective commercial biocontrol product against *F. graminearum*. This is evident in available patents registered, such as those for *Bacillus* species against FHB in cereals [117,118]. The widespread adoption of these patented products to control FHB can benefit organic farming with a healthier and more sustainable wheat product.

Massive screening of various *Bacillus* strains against a wide array of crop pathogens is still nonetheless necessary to identify new antagonistic species. Furthermore, the application of new tools and techniques for assessing the efficacy of biocontrol agents against crop pathogens can accelerate the discovery of new biocontrol strains of *Bacillus*. Equally important is the study of the mechanism of action of *Bacillus* against *F. graminearum*, which should be analyzed more accurately using the new tools of genome-wide studies and the sensitive and accurate platforms of metabolomics. High-resolution techniques of chromatography and mass spectrometry can make the detection of new antagonistic molecules possible even at traceable levels. Specifically, if explored extensively, *Bacillus* may replace in the control *F. graminearum* most of the current widely applied control agents, such as fungicides, and cultural practices which impact negatively on health and the environment.

## Figures and Tables

**Figure 1 toxins-11-00606-f001:**
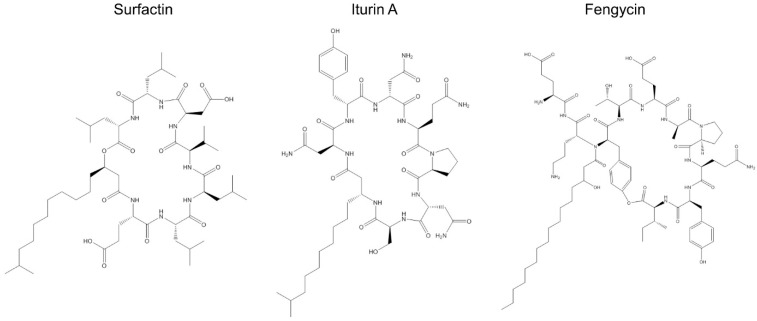
Congener structures of the cyclic lipopeptides; surfactin, iturin A, and fengycin. source: [61].

**Figure 2 toxins-11-00606-f002:**
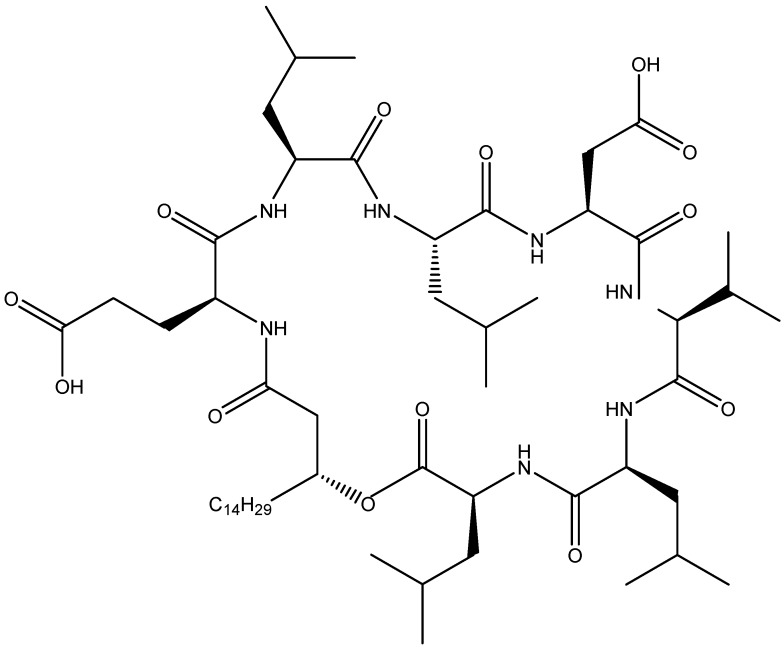
Structure of WH1fungin; source: [72].

**Figure 3 toxins-11-00606-f003:**
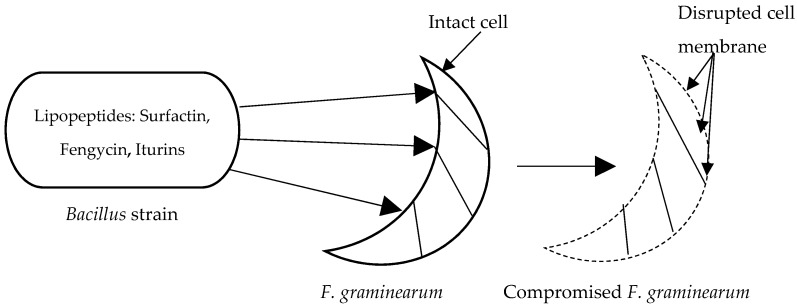
Graphical illustration of *Fusarium graminearum* cell disruption by *Bacillus*.

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
