# Peer review of "The Mode of Action of Bacillus Species against Fusarium graminearum, Tools for Investigation, and Future Prospects"

_toxins, 2019, doi:10.3390/toxins11100606_

Round 1
Reviewer 1 Report
Dear authors.
This work aims to familiarize readers with aspects on the use of Bacillus as a biological control agent for Fusarium graminearum.
F. graminearum causing Fusarium head blight and also other species of Fusarium are causing severe yield and grain quality loses. F. graminearum and F. oxysporum are ranked as top 4 and 5 fungal pathogen in molecular plant pathology by Dean et al. Moreover F. graminearum is well known for producing numerous mycotoxins, which decrease the quality of crops even more, while being potential threat to consumer health.
In this work authors focused on describing modes of action of Bacillus against F. graminearum.
I found this work very interesting, well designed and the said study field thoroughly explored.
I have some minor suggestions:
Please relate to the potential use of said modes of action and Bacillus spp. in organic/ecological farming. Organic farming is becoming more and more popular in recent years and fungal diseases are even more severe for it, than in traditional agriculture. I think that covering this topic may broaden reception of your work. Figure 1. Caption – Capitalize first letter. Line 149 – should be “agrastatin 1”Author Response
Incorporation of reviewers suggestions for Manuscript ID: toxins-577839
Response to Reviewer 1
We want to thank the reviewers for their constructive feedback on our manuscript, to improve its quality. All the suggestions and recommendations have been considered and addressed by the authors. Where applicable, new additions to the manuscript were highlighted in yellow and where applicable (strikethrough text). Details of each correction/rebuttal is provided in the table below:
Reviewer 1 |
|
Comments |
Response |
Please relate to the potential use of said modes of action and Bacillus spp. in organic/ecological farming. Organic farming is becoming more and more popular in recent years and fungal diseases are even more severe for it, than in traditional agriculture. I think that covering this topic may broaden reception of your work. |
This has been addressed and now in Section 5 (the first paragraph in red font). The added references have also been appended and also appear in red font. |
Figure 1. Caption – Capitalize first letter. |
This has been done. |
Line 149 – should be “agrastatin 1” |
This has been corrected. |
Reviewer 2 Report
The manuscript describes the present state-of-the-arts about about the employment of Bacillus spp. as suitable tool for the biocontrol of Fusarium graminearum and it also focus of the available methodologies for study of the physiological and molecular mechanism behind the antagonistic behaviour.
I think that this is an interesting manuscript worthy to be published in TOXINS after the following minor modifications:
Line 42. Modify the wording "...ADON and 15-ADON) reported..." as "...ADON and 15-ADON). These mycotoxins are reported..."
Line 43. delete the word "their": Add a full stop at the end of the line.
Lines 44-45. Modify the sentences as follow "However this effects vary from one animal species to the other and according to several factors as, trichothecene type, level and route of exposure. Taken together, the above evidences justify the needing for the biocontrol of F. graminearum in several foodstuffs.[12].
Line 61. Change the word "mode" in "mechanism".
Line 102. Change the wording "... This confirmed... " in "These findings confirmed".
Lines 105-106. Change the wording "compared to the controls. This was due to the presence of" in "since it showed to produce".
Lines 140-141. Change the wording "With the capabilities 139 of these strains to control F. graminearum" in "Because of this".
Lines 153-154. Modify congener as Congener.
Line 156. Change the wording "that have demonstrated antifungal" in "that have demonstrated to possess antifungal".
Line 160. Change the word "property" in "molecular structure".
Lines 170-171. Change the sentence "On F. graminearum, its effect can be culture conditions-dependent [17,19] with the concentration of iron among the determinants in producing surfactin [19]." in "This effect on F. graminearum can be culture conditions-dependent [17,19] with the concentration of iron being the most important determinant [19]."
Line 196. Change the wording "fungal cells, compromising" as "fungal cells by compromising".
Lines 228-231. The Figure 3 is not informative and it should be deleted.
Lines 241-242. Change the wording "...F. graminearum. As useful as bioassays, which involve co-culturing of antagonistic microbes, are the actual antagonistic mechanisms remain unknown." as "...F. graminearum, whose antagonistic mechanism still remain not fully explained".
Lines 254-255. Change the wording ".scenarios, i.e., there is a candidate compound (suspect) under investigation." in ".scenarios. The first is when the presence of a specific compound responsible of the antagonistic effect is supposed".
Line 256. Change the wording "Alternatively, if no compound is suspected, in " Alternatively, if the presence of a specific molecule is not supposed".
Line 259. Change the wording "uncovered through" in "made available by".
Lines 270-271. Delete the wording "Specifically, against F. graminearum,".
Line 278. Change the wording "...compounds still proving the value of genomic..." in "...compounds. These findings proved the value of genomic...".
Line 280. Change the word "utilize" in "have utilized".
Line 288. Change the wording "... substances against..." in " substances able to act against...".
Line 308-309. Change the wording "...The success of Bacillus species as biocontrol agents against Fusarium graminearum is opening new avenues for the continued exploitation of Bacillus in crop protection." in "...The evidence that Bacillus species can act as biocontrol agents against Fusarium graminearum encourages the exploitation of Bacillus in crop protection."
Line 310. Change the wording "..uncover new opportunities" in "identify new antagonistic species".
Line 313. Change the word "can" in "should".
Line 320-321. Change the wording "...This review is a unique reflection of the mode of action of the biological control bacterium Bacillus against F. graminearum." in "...This paper has aimed to review the actual knowledge on the mode of action of the biological control bacterium Bacillus against F. graminearum.".
Lines 323-334. These last sentences (starting from the wording "The production and use....) are just a useless summary of what discussed in the previous sections. They must be removed and substituted with one or more sentences discussing the future perspectives of the employment of Bacillus spp. in the biological fight against F. graminearum.
Author Response
Incorporation of reviewers suggestions for Manuscript ID: toxins-577839
Response to Reviewer 2
We want to thank the reviewers for their constructive feedback on our manuscript, to improve its quality. All the suggestions and recommendations have been considered and addressed by the authors. Where applicable, new additions to the manuscript were highlighted in yellow and where applicable (strikethrough text). Details of each correction/rebuttal is provided in the table below:
Reviewer 2 |
|
Comment(s) |
Response |
Line 42. Modify the wording "...ADON and 15-ADON) reported..." as "...ADON and 15-ADON). These mycotoxins are reported..." |
This has been corrected. |
Line 43. delete the word "their": Add a full stop at the end of the line. |
This has been adopted. |
Lines 44-45. Modify the sentences as follow "However this effects vary from one animal species to the other and according to several factors as, trichothecene type, level and route of exposure. Taken together, the above evidences justify the needing for the biocontrol of F. graminearum in several foodstuffs.[12]. |
This has been modified and adopted. |
Line 61. Change the word "mode" in "mechanism". |
This has been corrected. |
Line 102. Change the wording "... This confirmed... " in "These findings confirmed". |
This has been adopted. |
Lines 105-106. Change the wording "compared to the controls. This was due to the presence of" in "since it showed to produce". |
This has been addressed. |
Lines 140-141. Change the wording "With the capabilities 139 of these strains to control F. graminearum" in "Because of this". |
This has been adopted. |
Lines 153-154. Modify congener as Congener. |
This has been adopted. |
Line 156. Change the wording "that have demonstrated antifungal" in "that have demonstrated to possess antifungal" |
This has been modified and adopted. |
Line 160. Change the word "property" in "molecular structure". |
This has been adopted. |
Lines 170-171. Change the sentence "On F. graminearum, its effect can be culture conditions-dependent [17,19] with the concentration of iron among the determinants in producing surfactin [19]." in "This effect on F. graminearum can be culture conditions-dependent [17,19] with the concentration of iron being the most important determinant [19]." |
This has been adopted. |
Line 196. Change the wording "fungal cells, compromising" as "fungal cells by compromising". |
This has been adopted. |
Lines 228-231. The Figure 3 is not informative and it should be deleted. |
Figure 3 has been represented and much more informative. |
Lines 241-242. Change the wording "...F. graminearum. As useful as bioassays, which involve co-culturing of antagonistic microbes, are the actual antagonistic mechanisms remain unknown." as "...F. graminearum, whose antagonistic mechanism still remain not fully explained". |
Modified and adopted. |
Lines 254-255. Change the wording ".scenarios, i.e., there is a candidate compound (suspect) under investigation." in ".scenarios. The first is when the presence of a specific compound responsible of the antagonistic effect is supposed". |
Adopted. |
Line 256. Change the wording "Alternatively, if no compound is suspected, in " Alternatively, if the presence of a specific molecule is not supposed". |
Modified and adopted. |
Line 259. Change the wording "uncovered through" in "made available by". |
Adopted. |
Lines 270-271. Delete the wording "Specifically, against F. graminearum,". |
Adopted. |
Line 278. Change the wording "...compounds still proving the value of genomic..." in "...compounds. These findings proved the value of genomic...". |
Adopted. |
Line 280. Change the word "utilize" in "have utilized". |
Adopted. |
Line 288. Change the wording "... substances against..." in " substances able to act against...". |
Adopted. |
Line 308-309. Change the wording "...The success of Bacillus species as biocontrol agents against Fusarium graminearum is opening new avenues for the continued exploitation of Bacillus in crop protection." in "...The evidence that Bacillus species can act as biocontrol agents against Fusarium graminearum encourages the exploitation of Bacillus in crop protection." |
Adopted. |
Line 310. Change the wording "..uncover new opportunities" in "identify new antagonistic species". |
Adopted. |
Line 313. Change the word "can" in "should". |
Adopted. |
Line 320-321. Change the wording "...This review is a unique reflection of the mode of action of the biological control bacterium Bacillus against F. graminearum." in "...This paper has aimed to review the actual knowledge on the mode of action of the biological control bacterium Bacillus against F. graminearum.". |
This statement has been deleted. |
Lines 323-334. These last sentences (starting from the wording "The production and use....) are just a useless summary of what discussed in the previous sections. They must be removed and substituted with one or more sentences discussing the future perspectives of the employment of Bacillus spp. in the biological fight against F. graminearum. |
This has been adopted. |
This manuscript is a resubmission of an earlier submission. The following is a list of the peer review reports and author responses from that submission.